# SRL: Scaling Distributed Reinforcement Learning to Over Ten Thousand Cores

**Zhiyu Mei**[* 12♭], **Wei Fu**[* 12♯], **Jiaxuan Gao**[12], **Guangju Wang**[3], **Huanchen Zhang**[12], **Yi Wu**[123♮]

[1] IIIS, Tsinghua University, [2] Shanghai Qi Zhi Institute, [3] OpenPsi Inc.

♭ meizy20@mails.tsinghua.edu.cn, ♯ fuwth17@gmail.com,

♮ jxwuyi@gmail.com

## Abstract

The ever-growing complexity of reinforcement learning (RL) tasks demands a distributed system to efficiently generate and process a massive amount of data. However, existing open-source libraries suffer from various limitations, which impede their practical use in challenging scenarios where large-scale training is necessary. In this paper, we present a novel abstraction on the dataflows of RL training, which unifies diverse RL training applications into a general framework. Following this abstraction, we develop a scalable, efficient, and extensible distributed RL system called **_ReaLly Scalable RL_** (**SRL**), which allows efficient and massively parallelized training and easy development of customized algorithms. Our evaluation shows that SRL outperforms existing academic libraries, reaching at most **_21x_** higher training throughput in a distributed setting. On learning performance, beyond performing and scaling well on common RL benchmarks with different RL algorithms, SRL can reproduce the same solution in the challenging hide-and-seek environment as reported by OpenAI with up to 5x speedup in wall-clock time. Notably, SRL is the first in the academic community to perform RL experiments at a large scale with over 15k CPU cores. SRL source code is available at: https://github.com/openpsi-project/srl.

## 1 Introduction

Reinforcement Learning (RL) has been a popular paradigm leading to a lot of AI breakthroughs, including defeating world champions in various competitive games (Silver et al., 2016; Berner et al., 2019; Vinyals et al., 2019; Pérolat et al., 2022), controlling a tokamak nuclear fusion reactor (Degrave et al., 2022), discovering novel algorithms (Fawzi et al., 2022), and creating intelligent robots (Liu et al.,

Table 1: Capabilities of open-source distributed RL systems.

|  | Multi-Node Training | Remote GPU Inference | Custom RL Algo. | Custom DataFlow |
|---|---|---|---|---|
| RLlib | ✗ | ✗ | ✓ | ✗ |
| SeedRL | ✗ | ✓ | ✗ | ✗ |
| ACME | ✗ | ✗ | ✓ | ✗ |
| MSRL | ✗ | ✗ | ✓ | ✓ |
| SRL (ours) | ✓ | ✓ | ✓ | ✓ |

2022). As RL tasks get increasingly more complex, training a strong neural network policy requires millions to billions of trajectories. Generating the data sequentially would take hundreds or even thousands of years. Therefore, building a system that can parallelize the data collection process and perform efficient RL training over massive trajectories becomes a fundamental requirement for applying RL to real-world applications.

Numerous open-source libraries or frameworks are available to facilitate efficient RL training. However, we have identified several limitations in these systems that hinder their abilities to train RL agents efficiently in various scenarios, as we will show in Sec. 2.2. Such limitations render open-source libraries insufficient for supporting large-scale RL training applications like AlphaStar (Vinyals et al., 2019) for StarCraft II or OpenAI Five (Berner et al., 2019) for Dota 2. Unfortunately, the proprietary systems used by OpenAI and DeepMind have not been open-sourced, and the architectural and engineering details have not been disclosed to the research community. As a result, it remains

---

*Equal Contribution.

largely unknown how to develop a system capable of scaling well enough to support RL training in complex environments and maximizing hardware efficiency under resource constraints.

In this paper, we present a general abstraction of the dataflows of RL training. This abstraction effectively unifies training tasks in diverse circumstances into a simple framework. At a high level, we introduce the notion of *workers* to represent computational and data management components, each of which hosts distinct *task handlers* such as environments or RL algorithms. Workers are interconnected by *streams* and supported by *services*. Based on such an abstraction, we propose SRL (*ReaLly Scalable RL*), a scalable, efficient, and extensible distributed RL system. Capabilities of SRL compared to exitsing systems are shown in Table 1. SRL encompasses three primary types of workers to decouple major computations in RL training, which allows SRL to allocate suitable computing resources (CPUs or GPUs with varying computation powers) based on the requirements of each task in a cluster with heterogeneous hardware. Moreover, SRL is highly adaptable and extensible. In addition to a set of built-in implementations of workers and algorithms, SRL provides a general set of APIs that allow users to develop new algorithms and system components easily.

Our evaluation focused on two key metrics: *training throughput* and *learning performance*. We first compared the training throughput of SRL to various open-source libraries, showing the superior performance of SRL in both local and distributed settings. While open-source libraries generally fail to scale up to a large cluster, we re-implement their system architectures within SRL and compare them with the novel architecture in SRL. Such evaluation shows the extraordinary efficiency and scalibility of SRL's decoupled architecture. As for learning performance, we implement several popular RL algorithms and evaluate them at various scales on common RL benchmarks. These algorithms can obtain a reasonably high performance after 8 hours, as well as showing insightful scaling trends as the experiment scale increases. Finally, we evaluated SRL in the challenging hide-and-seek environment (Baker et al., 2019) and found that SRL is able to reproduce the same solution quality as *Rapid* (Berner et al., 2019), OpenAI's production system, while achieving a 3x speedup with CPU inference and a 5x speedup with GPU inference. SRL was also able to solve a more challenging variant of HnS with over 15K CPU cores. The training process gets substantially accelerated with an increasing amount of computation.

We make three primary contributions in this paper. First, we introduce a general abstraction on RL dataflows, which facilitates massive parallelism and efficient resource utilization under various resource settings. Second, based on this abstraction, we develop SRL, a scalable, efficient, and extensible distributed RL system. SRL features high-throughput training on both local machines and very large clusters, and flexible APIs that allows easy development of customized algorithms and system components. Third, we evaluate the performance of SRL extensively across a wide range of RL testbeds and algorithms in a large scale with up to 15k CPU cores. SRL shows superior scalability and efficiency compared to existing academic systems as well as OpenAI's production system.

## 2    BACKGROUND & MOTIVATION

### 2.1    REINFORCEMENT LEARNING SYSTEM

We identify three major computation components in a typical RL system: **Environment simulation** produces observations and rewards based on actions. This computation is typically performed using external black-box programs on CPUs. **Policy inference** produces actions from observations via neural network inference.**Training** executes gradient descent iterations with the collected trajectories to improve the policy. For computations on neural networks, the system can use either CPU or GPU devices, although there can be a clear performance advantage when adopting GPUs. Vanilla RL implementations tightly couple the above computations: at each training iteration, it first conducts environment simulation and policy inference alternatively to collect trajectories, and then performs training with the data. This vanilla implementation is far from efficient because only one of the major computations can be executed at a given time.

There have been numerous works aimed at developing RL systems or libraries only focusing on limited purposes and scales (Nair et al., 2015; Caspi et al., 2017; Hafner et al., 2017; Gauci et al., 2018; Castro et al., 2018; Fan et al., 2018; Stooke & Abbeel, 2018; 2019; Pardo, 2020; Zhi et al., 2020; Petrenko et al., 2020). OpenAI and DeepMind have developed industrial-scale distributed RL systems for training complex agents (Berner et al., 2019; Vinyals et al., 2019; Hessel et al., 2021).

Figure 1: IMPALA-style (left) and SEED-style (right) architecture implementations on a cluster with GPU nodes. The former merges environment simulation and policy inference in a single CPU/GPU node, while the latter merges policy inference and training on centralized GPU node. Note that in SEED-style, GPU nodes running environment simulation rely on the training GPU node for policy inference, while in IMPALA-style, they rely on local CPU/GPU for policy inference.

Unfortunately, they never open-source their systems or explain the architecture or engineering details. The open-source versions (Dhariwal et al., 2017; Küttler et al., 2019) of these systems are typically limited to small-scale settings and benchmarking purposes.

Other open-source and academic RL systems that enables distributed training typically follows two types of architectures. The first one is *IMPALA-style* (Espeholt et al., 2018) architecture (Fig. 1 left), which tightly couples environment simulation and policy inference. The coupled environment-policy loop can be parallelized to speed up training sample generation. This architecture is widely adopted by existing systems. Among them, RLlib (Liang et al., 2017), implemented with Ray (Moritz et al., 2017), is probably a most popular one. RLlib also exploits a programming model named RLlibFlow (Liang et al., 2020) to reduce coding efforts. ACME (Hoffman et al., 2020) is a framework with similar architecture to RLlib, based on a customized communication library Launchpad (Yang et al., 2021). Other similar works that share the architecture include ElegantRL (Liu et al., 2021) and TLeague (Sun et al., 2020). The other one, the *SEED-style* architecture (Fig. 1 right) proposed by SeedRL (Espeholt et al., 2019), features centralized policy inference and training on its single TPU/GPU "learner". SeedRL is originally specialized for RL training on TPUs, while remains compatible with GPU devices. MSRL (Zhu et al., 2023) is a concurrent work to us that supports both IMPALA- and SEED-style architectures. It transforms RL algorithm implementations into executable code fragments at runtime. Then these fragments are parallelized and executed following a pre-determined scheduling plan.

## 2.2 LIMITATIONS OF EXISTING SYSTEMS

Based on our research and evaluation, we have found that existing systems have several common limitations in both designs and implementations.

**Limitation 1: Low Resource Efficiency.** The two architectures adopted by existing systems have made unnecessary assumptions in available computation resources. Consequently, their architectures tightly *couple multiple computational components* and allocate them onto devices located on the same node, which could be highly inefficient in a customized cluster. The *IMPALA-style* architecture, represented by RLlib and ACME, assumes only local devices are available for policy inference to produce actions for environment simulations. The variation of inference requirements may easily cause local computing resources (CPUs or GPUs) to be idle or overloaded and lead to significant resource waste. The *SEED-style* architecture primarily assumes that multi-core TPU is available for training and policy inference. In the case of GPUs instead of a multi-core TPU, however, its computing power may struggle to handle both inference and training. Note that although MSRL implements both of the two architectures, it does not provide additional solutions to overcome their limitations caused by coupling.

**Limitation 2: Simplistic Implementations with Inadequate Optimizations.** Existing open-source systems and libraries have limited supports for multi-node training and performance optimizations. Specifically, RLlib and ACME only support multi-GPU training on a single node, while SeedRL can only run on a single trainer. Although the original design of MSRL include multi-learner options, only single-learner training is available in their open-source codebase. Moreover, the implementations of their components are usually single-threaded without overlapping I/O heavy operations and computation or fully utilizing idle time caused by data dependencies. Consequently, these systems lead to poor training throughput, especially in large-scale scenarios.

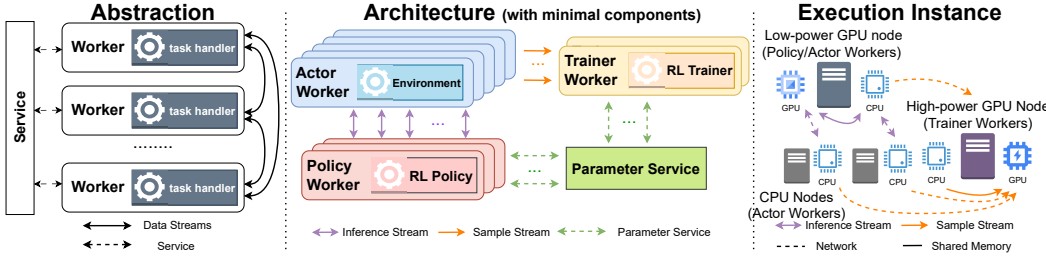

Figure 2: **(left)** In SRL abstraction, *workers* host *task handlers* to execute computing tasks. Workers are connected by *data streams* and supported by *services*. **(middle)** Based on the abstraction, the architecture for a typical RL workflow in SRL incorporates 3 types of core workers, 2 types of streams and the parameter services. **(right)** In an execution instance of SRL, workers are assigned appropriate resources on heterogeneous nodes in a distributed cluster. Data streams exploit fastest available communication substrates to ensure high-throughput data transmission.

**Limitation 3: Coupled Algorithm and System Interfaces.** In most open-source systems, the algorithm implementation is closely intertwined with their system APIs. In order to customize the algorithm, the users are forced to understand and modify codes related to system execution. Additionally, these systems assume a classic RL workflow that strictly follows the three major RL computation components mentioned in Sec. 2.1. As a result, they lack an interface that enables the expansion of system components, which restricts their compatibility with complex RL algorithms (such as Re-analyzed MuZero (Schrittwieser et al., 2019)) that require addtional computations.

## 3 SYSTEM DESIGN & ARCHITECTURE

In this section, we present the design and architecture of SRL, addressing all previously mentioned limitations. Sec. 3.1 introduces a general abstraction for RL training dataflows. Base on this abstraction, we introduce the architecture of SRL. For a clear presentation, the architecture demonstrated here only contains minimal components that is sufficient to support the most popular RL algorithm PPO (Schulman et al., 2017). The workflow of PPO is already introduced in Sec. 2.1. Next, Sec. 3.2 and Sec. 3.3 describes detailed designs, implementations and optimizations in core components of SRL, which reveals the factors that contribute to SRL's high performance. Finally, in Sec. 3.4, we introduce the extensible and user-friendly interfaces in SRL, showing its capability to be easily extended to support RL algorithms that require a more complicated system design.

### 3.1 HIGH-LEVEL DESIGN OF SRL

In order to achieve high resource efficiency in various computation resource settings, we propose a general abstraction to express RL training dataflows. As Fig. 2 left shows, the abstraction is composed of multiple *workers* that host distinct *task handlers* that execute computing tasks. These workers are connected via data *streams* and supported by background *services*. Unlike previous designs, *all* workers in SRL can be *independently scheduled and distributed* across multiple machines with heterogeneous resources, allowing massive parallelism and efficient resource utilization under various resource settings from single machines to large clusters. The data streams and background services follows different patterns depending on the requirement of various RL algorithms. It is also worth mentioning that the clean abstraction of SRL enables fine-grained optimization within each type of workers, which brings significant improvements in performance and scalability compared to existing libraries, even when adopting the same architecture.

Based on this simple abstraction, we develop the architecture of SRL as illustrated in Fig. 2 middle. We would like to emphasize that the architecture introduced here only contains minimal components that supports a typical RL workflow. This architecture incorporates three core types of workers: *actor worker*, *policy worker*, and *trainer worker*, which are responsible for the three pivotal workloads in RL training tasks. Data transmission between workers are primarily subsumed into two patterns and handled by *inference streams* and *sample streams*. Other communications are managed by services, for example, *parameter service* for parameter synchronization. Fig. 2 right demonstrates an execution instance of SRL in a distributed scenario. The workers are instantiated as processes on heterogeneous nodes in a cluster. Each worker is assigned an appropriate amount of computing resources based on

its task. Data streams exploit networking sockets or shared memory blocks as their communication substrates, facilitating high-throughput data transmission. As a result, the architecture of SRL not only unifies existing local and distributed architectures, but also enables efficient resource allocation and data communication for scenarios beyond the capabilities of existing architectures.

## 3.2 SYSTEM COMPONENTS & IMPLEMENTATION

In this section, we will describe the working logic and implementation of each system component.

**Actor workers** handle execution of black-box environment programs. Before every environment step, each actor worker sends out the observation and requests an action from policy workers to continue to the next step. Actor workers are inherently designed to be *multi-agent*, such that (1) different actor workers can host different number of agents, (2) agents can asynchronously step though the environment (skip if the environment returns `None`), and (3) agents can apply different policies by connecting to different streams (see App. A for an example use case). This flexible design enables SRL to deal with more complex and realistic application scenarios.

**Policy workers** provide batched policy inference services for actor workers. Policy workers flush inference requests, compute forward passes with batched observations, and respond to them with output actions. Jobs on the CPU and GPU are separated into two threads. Our implementation of policy workers also supports a local mode using CPU cores, which we call *inline inference*. In this case, the inference stream module will ensure direct data transmission between an actor and its associated local policy worker with proper batching without redundant data transmission, which can be preferred when no GPU devices are available.

**Trainer worker** consumes samples and updates policy model. Each trainer worker is embedded with a buffer. At each step, a trainer worker aggregates a batch of trajectories from the buffer and loads them into GPU for gradient updates. Jobs on the CPU and GPU are separated into two threads. Trainer workers use PyTorch DistributedDataParallel (DDP) (Li et al., 2020) for data parallel training.

**Sample streams and inference streams.** In SRL, we identify two primitive types of data transmissions. The first is exchanging observations and actions between actor and policy workers, while the other is pushing samples from actor to trainer workers. In SRL, we developed inference streams and sample streams to handle these two patterns, respectively. In one experiment, multiple instances of streams can be instantiated to establish independent and perhaps overlapped communications, which can be preferred in applications like multi-agent RL and population-based training.

**Parameter server** is the intermediate station for synchronizing policy models between policy and trainer workers. Trainer workers will push a newer version of policy models to the parameter server and policy workers will pull the model from it regularly. SRL adopts an NFS (Sandberg et al., 1985) implementation for the parameter server, which has enough throughput according to our experience.

**Controller and scheduler.** Upon launching an experiment, SRL submits a controller and all workers to the cluster via a scheduler. The controller is responsible for monitoring and managing the lifetime of workers. The scheduler supports a rich set of configuration options to do customized resource allocation, e.g., allocating trainer and policy workers to the same node with remote actors. Workers allocated to the same node will automatically establish shared-memory stream connection. The scheduler design prevents resource waste when some workers cannot fully utilize the power of a GPU and maximally reduces data transmission overhead in the cluster.

## 3.3 PERFORMANCE OPTIMIZATION

In this section, we introduce two optimizations that significantly contribute to the performance of SRL. Other optimizations are discussed in App. B and ablation studies are illustrated in App. C.

**Environment Ring.** In a trivial implementation of actor workers, CPU cores will be periodically idle when waiting for the next actions. To fully utilize the CPU resources, an actor worker in SRL maintains multiple environment instances and executes them sequentially in an *"environment ring"*. When an environment instance finishes simulation and starts waiting for an action, the actor will immediately switch to the next one. With a proper ring size, we can ensure that when simulating an environment instance, the required action is always ready. Environment rings substantially eliminate idle time for actors, and thus greatly increase data generation efficiency for actor workers.

**Trainer Pre-fetching.** A typical working cycle for a trainer worker consists of three steps: storing received data in the buffer, loading data into GPU, and computing gradients for model updates. The first two steps are I/O heavy operations. Overlapping these steps with the third step can lead to higher sample throughput than executing them sequentially on the trainer worker. While receiving and storing data into the buffer could be simply implemented into a separate thread, model updates on a sample batch require data loading as a dependency. To overlap computing with I/O operations, we use a *pre-fetching* technique. To implement this technique, we reserve GPU memory for additional batches of training samples. When training starts, one sample batch is fetched into GPU memory. Then, while the GPU computes the gradient on this sample batch, another sample batch is pre-fetched into the other memory block simultaneously. In this way, all three steps in the working cycle will be executed in parallel threads, which further improves the performance of SRL.

## 3.4 USER-FRIENDLY AND EXTENSIBLE DESIGNS

```python
class MyDQNPolicy:
    def __init__(self, **kwargs):
        # Can be any neural networks.
        self.net = MyNet(**kwargs)
        ...
    def rollout(self, request: Dict[Tensor]):
        # This method is used by policy workers.
        # "request" contains obs, policy state, ..
        ...
    def analyze(self, sample: Dict[Tensor]):
        # This method is used by the RL algorithm.
        # "sample" contains obs, act, rew, ..
        ...
class MyDQNAlgorithm:
    def step(self, sample: Dict[Tensor]):
        # The API called by trainer workers.
        q, q_target = self.policy.analyze(sample)
        loss = mse_loss(q, q_target)
        ... # Gradient descent step.
        return {'loss': float(loss)}
```

Code 1: Example of a simplified DQN policy.

```python
class Worker:
    def configure(self, config):
        self._configure(config)
        ...
    def run(self):
        while not self.__exiting:
            r = self._poll()
            # RPC requests from controller
            self._server.handle_requests()
class BufferWorker(Worker):
    def _configure(self, cfg: api.config.BufferWorker):
        ... # Init streams and the policy.
    def _poll(self):
        # Connected with actor worker.
        x, cnt = self.up_stream.consume_to(self.buf)
        # The "task handler".
        y = self.policy.reanalyze(self.buf.get())
        # Connected with trainer worker.
        self.down_stream.post(y)
        return PollResult(sample_count=cnt, batch_count=1)
```

Code 2: An simplified implementation of a customized buffer worker for data reanalysis.

In addition to its high scalability and efficiency, SRL offers user-friendly and extensible interfaces, allowing for the development and execution of customized environments, policies and complex algorithms within its framework.

Policies and algorithms in SRL are separated from the system design, allowing users to develop new variants without using any system-related interfaces. In Code 1, we present an example of a Deep Q-Network (Mnih et al., 2013) policy. To write a new policy, users only need to write a policy class that defines the policy's behavior during data generation and training, and an algorithm file that specifies how to compute the scalar loss given data obtained from the policy. These files are independent of the system component implementations, allowing users to focus on algorithm development rather than execution details.

Additionally, SRL offers interfaces for both new workers and data streams, facilitating complete customization of dataflow and algorithmic modules. For example, to implement Re-analyzed MuZero, an additional module that periodically reprocesses generated samples is required. Code 2 shows a simplified example of `BufferWorker` that implements such a module. Although this module does not fit into the primary computation components of SRL, users can easily create a self-defined worker-type by inheriting the base `Worker`, and define the underlying communication with data streams. This worker-based customization further facilitates the development of complex RL training routines within our framework. More examples and details can be found in App. A.

## 4 EXPERIMENTS

Our evaluation focuses on two key metrics: *training throughput* and *learning performance*. Training throughput refers to the rate at which a system can process sample frames per second (FPS) for gradient updates, while learning performance measures the wall-clock time required to generate the

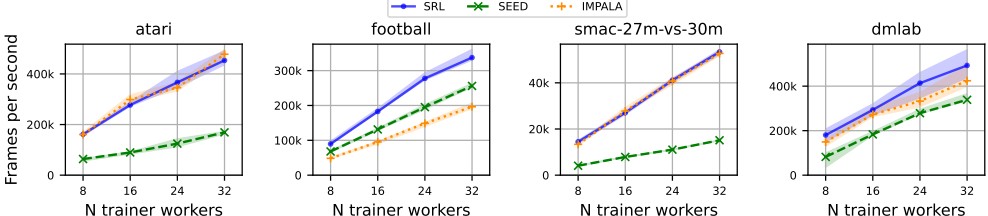

Figure 3: Training FPS of SRL and baselines on a single machine. SeedRL with 32 and 64 CPU cores results in GPU out-of-memory. SeedRL and Rlpyt do not support multi-agent environments.

Figure 4: Training FPS of three different architectures (SRL, IMPALA, SEED) implemented in SRL in a large-scale cluster, using up to 32 A100 GPU, 64 RTX 3090 GPU and 12800 CPU cores.

optimal solution or the reward obtained after training for a fixed amount of time. Full experiment details (including hardware resources, training parameters, etc.) are listed in App. D.

In terms of training throughput, we compare the end-to-end training FPS of SRL with open-source libraries in both single-machine and medium-scale distributed settings. Baselines generally fail to scale up to a large-scale cluster. To perform large-scale evaluation, we re-implement baseline architectures within SRL via adopting different scheduling configurations and compare them against SRL's fully decoupled architecture. These experiments demonstrate the super-high efficiency and scalability of SRL from academic use cases to large-scale production scenarios.

As for learning performance, we implement several popular RL algorithms within SRL and evaluate them at various scales on common RL benchmarks. Through these experiments, we validate the capability of SRL to unify different RL workflows, as well as presenting insightful scaling trends of representative RL algorithms. Finally, we focus on a realistic and challenging environment, hide-and-seek (HnS), to show the applicability of SRL for addressing complicated real-world problems.

## 4.1 TRAINING THROUGHPUT

**Environments & Algorithm** We run experiments on Atari (Bellemare et al., 2012) (*Pong*), Google Research Football (gFootball) (Kurach et al., 2019) (*11_vs_11*), StarCraft Multi-Agent Challenge (SMAC) (Samvelyan et al., 2019) (27m_vs_30m), and DMLab (Beattie et al., 2016) (*watermaze*), each of which possesses distinct characteristics in terms of observation type, speed, memory, etc (see Table 3). We employ the widely-used Proximal Policy Optimization (PPO) (Schulman et al., 2017) as our primary algorithm choice. Each trajectory will be consumed only once.

**Comparison with Baselines** We choose Sample Factory (Petrenko et al., 2020), rlpyt (Stooke & Abbeel, 2019), and SeedRL (Espeholt et al., 2019) as baselines in the single-machine setting and RLlib (Liang et al., 2017) as baseline in the distributed setting. Fig. 3 and Table 2 show results in the single-machine and distributed setting respectively. SRL achieves the best overall performance in both settings. In the single-machine setting, SRL beats state-of-the-art systems specialized for this setting without sacrificing scalability and extensibility. In the distributed setting, SRL achieved 6.3x to 21.6x higher maximal performance compared with RLlib. We remark that trainer workers of SRL can effectively utilize more training samples in contrast to RLlib's single-endpoint multi-threaded trainer. Moreover, actor workers in SRL are capable of generating more training samples with GPU-accelerated inference and specialized performance optimizations. As for MSRL (Zhu et al., 2023), the IMPALA and PPO algorithm implementations in their open-source codebase [1] use NCCL as the communication backend. This results in GPU requirement for each environment simulation

---

[1]MSRL trainer code link.

Table 2: Training throughput with 8 A100 GPU trainers with distributed actors. # CPU Cores (peak): CPU cores used for training sample generation when trainers reaches peak performance.

|  | Atari | DMLab | gFootball | SMAC |
|---|---|---|---|---|
| SRL(FPS) | 644k | 741k | 89k | 17k |
| # Cores (Peak) | 800 | 1600 | 3200 | 1280 |
| SRL(FPS) | 150k | 658k | 19k | 5.0k |
| # Cores | 96 | 160 | 700 | 200 |
| RLlib(FPS) | 66k | 34k | 5.8k | 2.7k |
| # Cores (Peak) | 96 | 160 | 700 | 200 |

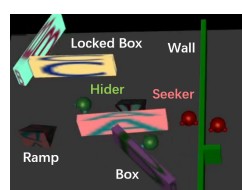

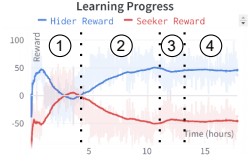

(a) HnS environment.

(b) Reward over time in HnS.

Figure 5: (a) A snapshot of HnS. (b) Rewards in HnS. Agent behavior evolves over four stages: running and chasing, box lock, ramp use, and ramp lock.

process, forbidding us to evaluate MSRL in the single-machine setting. Moreover, the open-source codebase does not involve multi-learner options, blocking us from experiments in our distributed setting. Hence, we evaluated against its performance described in paper. The detailed experiment settings and training throughput numbers are listed in App. D.2. The result shows that our system could reach 2.52x training FPS under the same experiment settings.

**Large-Scale Architecture Evaluation**  We implement IMPALA-style and SEED-style architectures in SRL, and evaluate them against the new decoupled architecture of SRL. Fig. 4 shows the full result. We progressively increased the allocation of computing resources from a quarter of the cluster (8 A100 GPUs and 3200 CPU cores) to the entire cluster. As the computing resources increase, SRL exhibits proportional performance improvement across all architectures, indicating favorable scalability. When employing a full-sized cluster, our novel architecture achieves a maximum improvement of 3.74x compared to SEED-style and 1.68x compared to IMPALA-style architecture.

## 4.2 LEARNING PERFORMANCE

**Common RL Benchmarks**  We implement four RL algorithms: (MA)PPO (Schulman et al., 2017; Yu et al., 2022), Apex-DQN (Horgan et al., 2018), and VDN (Sunehag et al., 2018). These algorithms cover a wide range of RL workflows range from on/off-policy to single/multi-agent. We select Atari as the single-agent benchmark suite and gFootball as the cooperative multi-agent counterpart. We train agents for a fixed amount of time (8 hours) and report the evaluation score of the final checkpoint over 100 episodes. For each algorithm, we choose a base configuration and multiply the batch size and the number of workers for larger scales. All experiments are submitted to our distributed cluster with GPU inference. See App. D for detailed configurations.

We show the aggregated task performance in terms of IQM (Agarwal et al., 2021) in Fig. 6. Per-task scores and hyperparameters are presented in App. D. Since training is terminated after 8 hours, the resulting score is not directly comparable to prior works or across different algorithms. We primarily draw two main conclusions from Fig. 6. First, SRL is able to support different RL algorithms with little engineering efforts. Within SRL, these algorithms can obtain a reasonably high performance after a short period of training. Second, the scaling trend shows that PPO acquires more benefits of the increased scale thanks to

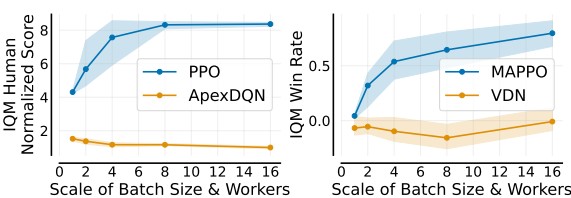

Figure 6: Algorithm performance in terms of IQM (Agarwal et al., 2021) human-normalized score in Atari-5 (Aitchison et al., 2023) and IQM win rate in gFootball academy scenarios after 8-hour training at various scales. Averaged over 5 seeds with 0.95 confidence interval shaded. PPO shows appealing scaling trend with more samples and larger batch sizes.

variance reduction, while the performance of Q-learning algorithms my drop because of less training steps under fixed time budget. This finding extends previous studies (Andrychowicz et al., 2021; Yu et al., 2022) and raises several open questions. We hope these questions could drive the community towards developing more advanced techniques for efficient RL training.

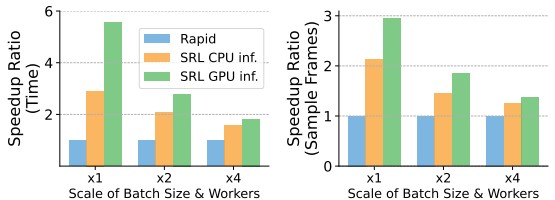
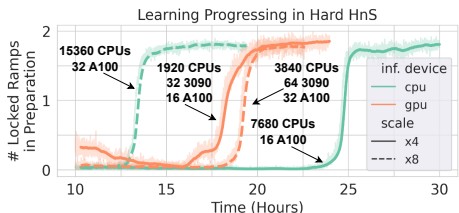

(a) Speedup ratio in HnS compared with Rapid.

(b) Learning progress in the hard setting of HnS.

Figure 7: (a) Speedup ratio to reach stage four in HnS compared with results reported in Baker et al. (2019). SRL achieves up to 5x acceleration. (b) Learning curves (i.e. the number of locked ramps in the preparation phase over time) in HnS with 2x map size at various scales.

**Reproducing Hide-and-Seek (HnS)**    We present the environmental details in App. D.5. Via training intelligent agents in HnS via PPO self-play, Baker et al. (2019) discovered four stages of emergent behaviors across learning. The reward across this learning process (reproduction within SRL) is shown in Fig. 4b. Due to the non-trivial agent-object interactions and the complexity of the task, observing such behavior evolution requires *an extremely large batch size, e.g. 320k interaction steps in a single batch*. We made our best efforts to run baseline systems in this environment, but *none* of them can handle training tasks on such a scale and reproduce the reported results. We run SRL with the same environment configuration, policy model, and PPO hyper-parameters as Baker et al. (2019). Since Rapid is not open-source, nor do the authors reveal how much computation resources they have utilized, we are only able to compare with the reported numbers in the original paper.

We conduct experiments in the distributed setting using both inline CPU inference (denoted as *CPU Inf.*) and remote GPU inference (denoted as *GPU Inf.*). In Fig. 7a, we present the acceleration ratio of training time and data volume required to achieve the ramp lock stage. Our results reveal that SRL is up to 3x faster than the Rapid system with the same architecture (*CPU Inf.*), while *GPU Inf.* can achieve up to 5x acceleration with further reduced time and environment interactions. We attribute the improvement in training efficiency to two reasons. First, our system design is more efficient and has a higher FPS than Rapid. Second, our flexible system design and fine-grained optimizations ensure that the efficiency of the RL algorithm is less affected by various system-related factors like network latency and out-of-date data, which leads to improved sample efficiency even with the same RL algorithm and hyperparameters.

**Solving *Harder* HnS with *over 15k* CPU cores.**    To explore the performance limit of SRL, we examine a more challenging HnS setting with double-sized playground area. This scenario has not been studied in previous works. Since the episode length is not changed, agents must learn to cooperatively search across a larger map for available objects and efficiently utilize the limited preparation phase. We run experiments with scale x4 and x8, and present learning progress over time in Fig. 7b. With the same computation resources (i.e., x4), SRL requires almost twice the time to achieve the ramp lock stage in this new task, highlighting its difficulty. Interestingly, while *GPU Inf.* x8 attained similar performance compared to the x4 scale, *CPU Inf.* x8 halved the required training time. We remark that *CPU Inf.* x8 utilizes up to 15000+ CPU cores and 32 A100 GPUs, which is much beyond the computation used in most prior research papers. This outcome not only highlights the exceptional scalability of SRL, but also demonstrates the potential benefits of utilizing RL training, specifically PPO, at a larger scale.

## 5    CONCLUSION

This paper presents a general abstraction of RL training dataflows. Based on the abstraction, we propose a scalable, efficient, and extensible RL system, SRL. The abstraction and architecture of SRL enables massively parallelized computation, efficient resource allocation, and fine-grained performance optimization, leading to high performances in both single-machine and large-scale scenarios. Also, the user-friendly and extensible interface of SRL facilitates customized environments, policies, algorithms, and system architectures for users. Our experiments demonstrate the remarkable training throughput and learning performance of SRL across a wide range of applications and algorithms.

ACKNOWLEDGMENTS

We extend our sincere gratitude to Qiwei Feng, who made significant contributions to establishing the basis of our code framework during the initial phase of this project. He also provided essential assistance in setting up the Slurm cluster.

We also extend our thanks to Zelai Xu, Hao Tang, Shusheng Xu, Chao Yu, Weilin Liu, and Yunfei Li for their contributions in implementing new features and adapting our system to various reinforcement learning applications. Their dedication greatly aided us in refining SRL and enhancing its functionality. The order of names listed does not imply a hierarchy of contribution.

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

## A  SRL INTERFACES & MORE CODE EXAMPLES

```python
ExperimentConfig(
    actors=[
        ActorWorker(
            env="remote_reward_compute_env",
            inference_streams=["pol_inf", "rwd_inf"],
            sample_streams=["rl_train", "null_stream"],
            agent_specs=[
                AgentSpec(
                    index_regex="0", # the first agent
                    # macthes to the first inf stream
                    inference_stream_idx=0,
                    sample_stream_idx=0,
                ),
                AgentSpec(
                    index_regex="1",
                    inference_stream_idx=1,
                    sample_stream_idx=1,
                )
            ],
        ) for _ in range(128)
    ],
    policies=[PolicyWorker(
        inference_stream="pol_inf",
        policy="rl_policy",
    ) for _ in range(4)] +
    [PolicyWorker(
        inference_stream="rwd_inf",
        policy="clip", # The large pretrained model.
    ) for _ in range(1)],
    trainers=[TrainerWorker(
        trainer="ppo",
        policy="rl_policy",
        sample_stream="rl_train",
    )],
)
```

Code 1: Example configuration for computing rewards in remote policy workers hosting large pre-trained models.

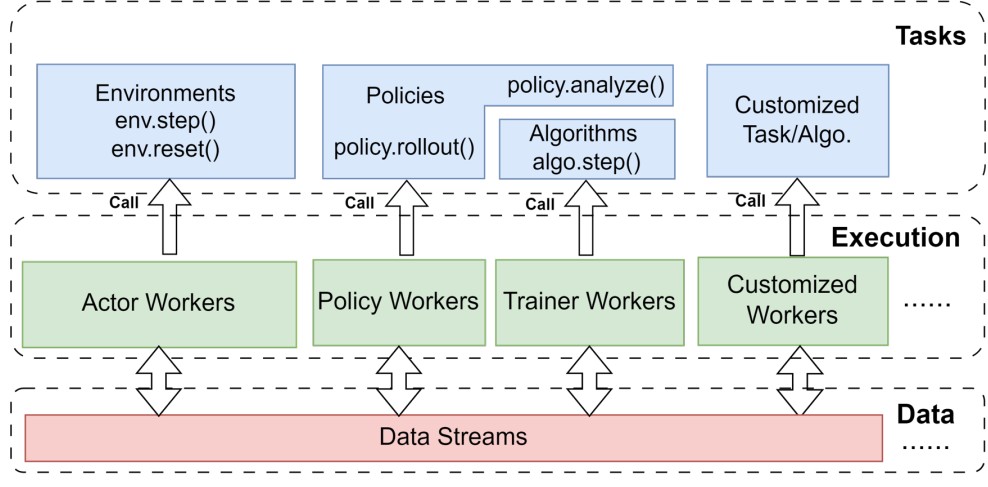

Figure 8: Graphical illustration of SRL interfaces.

Code 1 depicts a scenario where rewards returned by the environment must be computed using a large pre-trained neural network model rather than the program itself. With SRL, users can create an additional sentinel agent for reward computation. This agent is linked to another inference stream and policy worker, which hosts the reward model. After each environment step, real agents return `None` and the sentinel agent issues a reward computation inference request. After the reward is returned, the sentinel agent returns `NULL` and real agents return next-step observations and computed reward to advance environment simulation.

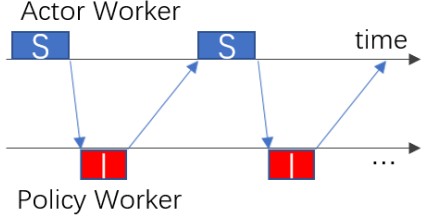

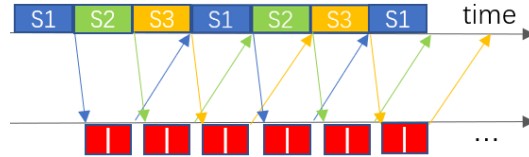

(a) Trivial actor worker with one env. instance.

(b) SRL actor worker with env. ring of size three (3 env. instances).

Figure 9: Timelines of actor workers. "S" marks env. steps. "I" marks inference steps. Arrows between timelines are observations and actions transmitted between actor worker and policy worker.

Fig. 8 illustrates the basic structure of the interfaces of SRL. The interface is separated into three levels, i.e. tasks level, execution level and data level. If users want to experiment on an existing algorithm on a pre-defined environment, they only need to modify experiment configuration, as depicted in Code 1. If users require customization of a new environment, policy or algorithm, they only need to implement APIs (e.g. `env.step()` and `algo.step()` ) in the tasks level. If the new algorithm requires additional dataflow or computation components, users can implement new workers and data streams easily with APIs provided in the execution and data levels.

Other concrete examples of configurations, algorithms and system components, and documentation of our full interfaces can be found in our (anonymous) code repository: `https://anonymous. 4open.science/r/srl-1E45/`

## B  PERFORMANCE OPTIMIZATIONS

In this section, we discuss effective optimizations techniques that contribute to the overall performance of SRL in details.

### B.1  ENVIRONMENT RINGS

In actor workers, environment simulation usually follows a two-stage loop: an actor simulates a black-box program by feeding in an action to receive an observation, and then waits until an action is received from the policy worker. In this process, the CPU will be periodically idle when the action is being processed by the policy worker (Fig. 9a). To fully utilize the CPU resources, an actor in SRL maintains multiple environment instances and executes each environment instance sequentially. We call this technique an *environment ring*. With an environment ring, when an environment instance finishes simulation and starts waiting for an action, the actor will immediately switch to the next environment instance. With a proper ring size, we can ensure that when simulating an environment instance, the required action is always ready (Fig. 9b). Environment rings substantially eliminate idle time for actors, and thus greatly increase data generation efficiency for actor workers. The optimal ring size usually depends on the speed of environment simulation, observation size, and policy model size.

### B.2  TRAINER DATA PRE-FETCHING

A typical working cycle for a trainer worker consists of three steps: storing received data in the buffer, loading data into GPU, and computing gradients for model updates. The first two steps, receiving and loading data, are I/O heavy operations. Overlapping these steps with the third step can lead to higher sample throughput than executing them sequentially on the trainer worker. While receiving and storing data into the buffer could be simply implemented into a separate thread, model updates on a sample batch require data loading as a dependency. To overlap computing the gradient for model update with data loading, we use a data pre-fetching technique.

To implement this technique, we reserve GPU memory for additional batches of training samples. When training starts, one sample batch is fetched into GPU memory. Then, while the GPU computes the gradient on this sample batch, another sample batch is pre-fetched into the other memory block

simultaneously. In this way, all three steps in the working cycle will be executed in parallel threads, which further improves SRL's training performance.

## B.3 SEPARATE CPU/GPU WORKER THREADS

For workers involving GPU computation, we leave CPU-heavy and I/O-heavy parts in the main worker poll thread, and initialize a new thread for GPU computation, including policy inference, policy re-analyze, and training. CPU-thread and GPU-thread communicate via queues. This design enables non-blocking logging, data transmission and parameter synchronization. Besides, it avoids additional data copy and context switching compared with the multi-process implementation.

## B.4 SHARED-MEMORY STREAM

For workers allocated on the same node, streams connecting these workers will automatically use shared memory to reduce communication overhead.

The shared-memory inference stream is implemented as two pinned memory blocks. The block size is set to be the total number of environments in connected actor workers. Each environment instance is associated with a specific block ID. After each environment step, the environment will write the returned observation into the observation block according to its own ID and mark the corresponding slot as ready. The policy worker will gather all ready slots in the block and perform batched policy inference, then the returned actions will be written to the action memory block with corresponding block IDs.

The shared-memory sample stream is a well-designed FIFO queue. The queue is composed of several out-of-order slots (i.e., they can be written and read in random orders), and each slot stores one piece of trajectory. When an actor worker pushes samples to the stream, the stream will allocate free slots and return them back to the actor worker for writing. The trainer worker randomly fetches from allocated slots, consumes the data once, and increase the usage counter of each fetched by one. Special care should be taken to manage the slot indices (e.g., whether it is being read/written or whether the sample reuse is exhausted). Besides, the shared-memory sample stream is zero-copy: data stored by actor workers will be directly loaded into GPU in the trainer worker without additional copying.

## B.5 DYNAMIC BATCHING

In the policy worker, we use a dynamic batch size to perform inference. The policy worker will wait until received inference requests exceeds the configured maximum batch size or the time exceeds the configured maximum idle time, then it will flush all accumulated requests to the GPU thread. Compared with fixed-batch-size inference, dynamic batching can automatically adjust the workload according to the ratio between actor and policy workers, and enable fault tolerance when some actor worker crashes due to unstable environment implementations.

## B.6 THREADED ENVIRONMENT RINGS

Most types of environments are computation-heavy simulators, but in some cases, environments hosted in actor workers can be internet brokers of remote environments. In other words, the physics engine or game simulator runs on the remote cloud and communicates observations, actions, and rewards with SRL actor workers. In this case, the standard environment ring will face several challenges:

1. Due to the fluctuation of internet connection, the variance of environment step is large. Environments stepped later may return earlier. Waiting for environments in the ring sequentially will cause large time waste.

2. The connection between brokers and the remote cloud may get lost. In a standard environment ring, this actor worker will idly wait for the response of the lost environment and also ignores all other healthy environments in the ring.

To address these challenges, we provide a threaded environment ring implementation. Environments are hosted in independent threads in actor workers. This implementation can largely benefit fault tolerance and increase the sample generation throughput for remote broker-based environments.

### B.7 DATA COMPRESSION

Transferring image-based observation or observations of multiple agents in streams is expensive. Since these data usually holds specific patterns (i.e., they are not nearly random numbers that are hard to compress, like the hidden states of a neural network), we optionally compress these data before sending them to the stream and decompress them on the receiver side. In our experiments, we compress all data sent via sockets with lz4.

## C ABLATION STUDIES

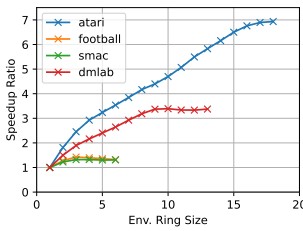 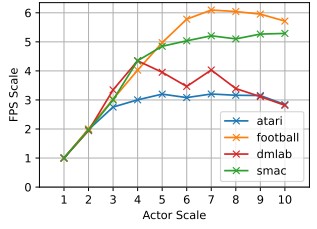 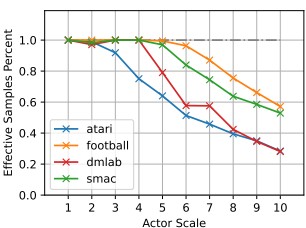

(a) Env. simulation speed up ratio of actor workers with different env. ring size.

(b) FPS scale on trainer worker with different numbers of actor workers.

(c) Utilized percentage of training samples produced by actor workers.

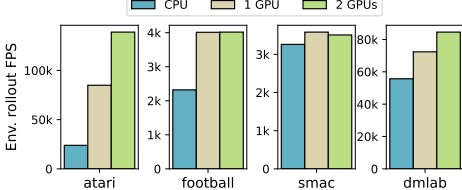 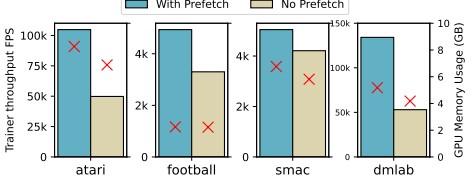

(d) Environment simulation FPS by 128 actor workers with difference inference devices.

(e) Training throughput on a Nvidia RTX 3090, with or without pre-fetching. Red cross marks GPU memory usage.

Figure 10: Ablation studies

This section presents a detailed analysis of the individual components of SRL. The purpose of these ablation studies is to offer practical recommendations for configuring SRL to achieve optimal performance in specific application scenarios. Our analysis encompasses fundamental systematic parameters and choices within SRL, such as the environment ring size, the employment of GPU inference, trainer worker pre-fetching, as well as the allocation of resources and ratios for each worker type.

### C.1 ENVIRONMENT RING

While the environment ring is not useful for increasing the throughput of inline CPU inference, a larger ring size can usually lead to a higher throughput of GPU inference. We measure the sample generation throughput by varying ring size inside a single actor worker with policy workers on a single GPU. Fig. 10a illustrates that as the ring size increases, the speedup ratio w.r.t. a single environment instance grows until it reaches a plateau. Note that a larger ring size also means greater memory consumption. We believe that the optimal ring size should trade off among three factors: memory consumption, environment simulation speed, and average inference latency. First, memory consumption can provide information about the maximum number of environments on each node. Second, a higher environment simulation speed usually indicates a larger optimal ring size — in our experiment, Atari has the smallest step time, resulting in the largest optimal ring size and the highest speedup ratio of around 7x. Users can propose a initial value of the ring size according to the previous two factors. Finally, since inference latency could be influenced by various factors in

practical situations (e.g. network condition, GPU capabilities), it is the most reliable to find the optimal ring size by experiments that simulate the practical policy worker and actor worker setups. We also note that the ring size should be the first thing to be determined for configuring SRL.

## C.2 GPU INFERENCE VS CPU INFERENCE

The next thing to determine is whether to use GPUs for policy inference and if so, how many GPUs should be used. From Fig. 4 we can see that if (1) the observation size is small, (2) the environment simulation speed is slow, and (3) the policy model is simple (e.g. a feed-forward network), inline CPU inference is usually sufficient. Otherwise, GPU inference can usually increase training throughput. Fig. 10d shows the sample generation throughput of 128 actor workers using different inference devices. Since DMLab and Atari have image-based observations and incorporate convolutional neural policies, they achieve their optimal throughput when using 2 GPUs, resulting in a 1.56x and 5.44x speedup ratio, respectively. Football and SMAC environments benefit less from GPU inference due to a slower pace and simpler policy models.

## C.3 MAXIMIZING TRAINER WORKER THROUGHPUT

Once the environment ring size and the ratio between actor and policy workers are determined, the sample generation speed of the system is fixed. The next concern is to maximize the number of samples consumed by trainer workers while minimizing wasted training samples. In the subsequent analysis, we focus on scenarios with a single trainer worker, while the number of workers can be scaled proportionally for larger-scale experiments. To demonstrate the relationship between trainer frames-per-second (FPS) throughput and the number of actor workers, we assessed the training throughput of a single trainer consuming training samples generated by varying numbers of actor workers. Fig. 10b illustrates that as the number of actors increases, the trainer FPS also rises until it reaches its maximum processing capacity. Beyond this point, the FPS might experience a slight decline due to network traffic congestion on the trainer worker. When a large number of actor workers are employed, redundant samples that cannot be utilized by the trainer are discarded. Fig. 10c presents the utilization percentage of training samples generated by all actor workers on the trainer worker. When an excessive number of actor workers are allocated to a trainer worker, the utilization percentage sharply decreases, indicating a waste of training samples. It is important to note that the outcome in the DMLab environment exhibits instability as the number of actor workers increases. This is primarily due to the lengthy and highly variable reset time of the environment, which is inherent to the original environment implementation and lies beyond the scope of our system optimizations.

## C.4 TRAINER WORKER PRE-FETCHING

In App. B, we introduced the concept of pre-fetching in trainer workers, which involves a trade-off between GPU memory utilization and training speed. This technique can be employed whenever there is a sufficient amount of available GPU memory, which is commonly the case in reinforcement learning contexts. To demonstrate the effectiveness of pre-fetching, we conduct an experiment comparing the training throughput and GPU memory utilization of a single trainer worker using an Nvidia 3090 GPU, both with and without the pre-fetching technique. Our results, presented in Fig. 10e, indicate that utilizing pre-fetching can lead to a 2-3x increase in throughput for Atari and DMLab, but less speedup on Google Football and SMAC. We believe that this discrepancy is due to the fact that Atari and DMLab are image-based environments, which generate samples at a faster rate with a larger size and possesses a more intricate policy model. Consequently, the trainer workers' capacity to consume samples and perform training operations can become a bottleneck. By increasing the number of samples that trainer workers can access through pre-fetching, the policy update process can be expedited, making it a more effective technique for such scenarios. On the other hand, GPU memory consumptions are increased by 100 MBytes to 1.5 GBytes, which is bearable compared to the total memory size (24 GBytes) of an Nvidia 3090 GPU.

Table 3: Characteristics of adopted environments.

| | Atari (*Pong*) | Deepmind Lab (*watermaze*) | gFootball (*11_vs_11*) | SMAC (*27m_vs_30m*) |
|---|---|---|---|---|
| Observation | Image | Image | Vector | Vector |
| # Controlled Agents | 1 | 1 | 22 | 27 |
| Third-Party Engine | No | No | Yes | Yes |
| Background Process | No | No | No | Yes |
| Pure Simulation FPS | 4800 | 300 | 75 | 35 |
| Memory (MBytes) | $\sim$100 | $\sim$250 | $\sim$170 | $\sim$1500 |

Table 4: Computing resources used in experiments in Sec. 4.1. AW: Actor Worker, PW: Policy Worker, TW: Trainer Worker. Each AW is allocated to one CPU core. 4 PWs are allocated to one Nvidia 3090 GPU. Each TW is allocated to one Nvidia A100 GPU.

| Env. | AW Per TW (Total) in Conf. 1&2 | AW Per TW (Total) in Conf. 3 | PW Per TW in Conf. 1 |
|---|---|---|---|
| Atari | 100 (3200) | 400 (12800) | 8 |
| SMAC | 160 (5120) | 200 (6400) | 4 |
| gFootball | 400 (12800) | 400 (12800) | 4 |
| DMLab | 200 (6400) | 400 (12800) | 8 |

# D EXPERIMENT DETAILS

## D.1 TRAINING THROUGHPUT

**Environments.** The four environments chosen in our throughput experiment are representative for common RL applications, considering simulation speed, observation types, memory occupation, number of agents and executable programs. Observation size for image-based environments are $84 \times 84$ greyscale images for Atari and $96 \times 72$ RGB images for DMLab. In Atari and DMLab environments, we adopt a traditional 4-frameskip setting. Other features of the four environemnts are listed in Table 3.

**Computing Resources.** Experiments are evaluated on two resource settings.

1. **Single-machine setting**: 32 physical CPU cores, 512 GB DRAM and 1 Nvidia 3090 GPU.
2. **Distributed cluster setting**: 4 nodes with 64 physical CPU cores and 8 Nvidia A100 GPUs + 64 nodes with 128 physical CPU cores and an Nvidia 3090 GPU. Each node in the cluster has 512GB DRAM, and they are connected by 100 Gbps intra-cluster network. Storage for the cluster is facilitated through NFS and parameter service, available on all nodes.

Note that All physical cores have hyperthreading enabled and count as 2 logical cores. In this section, if not emphasized, the term "CPU cores" will be referring to logical CPU cores. In comparison with baselines, results in Fig. 3 are evaluated on the single-machine setting, and results in Table 2 are evaluated on the cluster with 1 Nvidia A100 node and 16 Nvidia 3090 nodes. In large-scale experiments, to strike a balance between maximizing performance and minimizing computing resource waste, we carefully determine the number of workers of each type and the allocated resources based on our ablation study (refer to App. C). Table 4 presents the specific number of workers and computing resource utilization for each environment and architecture employed in this experiment.

**Training configurations.** In our baseline comparison experiments, we make our best effort to align training configurations for all baselines. Specifically, the model architectures and sizes are the same, and implemented with pytorch (Paszke et al., 2019) neural networks. We adopt specifications of policy models for each environments from prior works: Atari from DQN (Mnih et al., 2013), DMLab from SampleFactory (Petrenko et al., 2020), SMAC and GoogleFootball from MAPPO (Yu et al., 2022). From our experience, batch size and episode length does not evidently effect overall sample throughput of the system, except for the case when the batched samples are so small in size that inflict significant networking overheads (e.g. GoogleFootball). On the other hand, size of environment ring is crucial to the performance as illustrated in Fig. 10a. Detailed training configurations are listed in Table 5.

Table 5: Batch size (number of episodes per batch), max episode length and env ring size for 4 environments.

| Env. | Batch Size (per 3090 GPU) | Batch Size (per A100 GPU) | Max Episode Length | Env. Ring Size |
|---|---|---|---|---|
| Atari | 32 | 128 | 200 | 20 |
| SMAC | 32 | 128 | 100 | 2 |
| gFootball | 128 | 1024 | 100 | 4 |
| DMLab | 32 | 128 | 200 | 10 |

Table 6: Training throughput (FPS) of baseline comparison experiments in single-machine settings, see Fig. 3

| #CPU cores | 8 | 16 | 32 | 64 | 8 | 16 | 32 | 64 |
|---|---|---|---|---|---|---|---|---|
| | | SRL | | | | Sample Factory | | |
| Atari | 22183 | 40565 | 72681 | 124021 | 17248 | 31595 | 58831 | 96416 |
| DMLab | 9231 | 16614 | 28980 | 45974 | 8218 | 15448 | 27187 | 41994 |
| SMAC | 363 | 698 | 1234 | 2313 | 255 | 501 | 989 | 2001 |
| GFootball | 477 | 874 | 1474 | 2579 | 360 | 695 | 1382 | 2725 |
| | | SeedRL | | | | rlpyt | | |
| Atari | 21088 | 27918 | - | - | 9890 | 17406 | 18472 | 14816 |
| DMLab | 8933 | 14123 | - | - | 2902 | 4515 | 4226 | 7627 |

**Algorithm Selection.** We remark that on-policy methods are preferable for evaluating training throughput, as training speed is closely related to sample generation speed. Note that on-policy algorithms have a similar workload on the trainer worker side. Although other on-policy RL algorithms such as IMPALA (Espeholt et al., 2018) are also valid, we have chosen PPO as it is a highly popular RL algorithm for large-scale training and has demonstrated success in various RL applications (Berner et al., 2019; Baker et al., 2019; Ye et al., 2020).

**Baseline Selection.**

- ACME (Hoffman et al., 2020) shares a similar architecture as RLlib. However, the reported performance in Hoffman et al. (2020) is several times lower than RLlib in the same setting. The reported number also matches the results of our early experiment, so we omit ACME in the distributed setting.

- Although SeedRL (Espeholt et al., 2019) supports distributed execution, we find that samples generated with 32 CPU cores can easily overburden its single GPU "learner". Therefore, we only evaluated SeedRL in the single-machine setting.

**Result numbers.** The concrete numbers of experiment results illustrated in Fig. 3 and Fig. 4 are listed in Table 6 and Table 7.

## D.2 COMPARISON WITH REPORTED NUMBERS OF MSRL

In order to evaluate the performance of SRL against concurrent work MSRL, we conduct an experiment of training throughput in the setting of Sec. 6.2 Fig. 6(a) in MSRL (Zhu et al., 2023). In MSRL paper, they have evaluated PPO algorithm with 320 Mujoco Halfcheetah environments and 24 Nvidia V100 GPUs for inference. The result shows that MSRL can finish one episode of 1000 steps in 3.85s and 83116 FPS. In our experiment, SRL exploits the same numbers of CPU cores and environment instances, and 4 Nvidia 3090 GPUs for inference. The result show that the overall training throughput reaches 210165 FPS. The experiment details are obtained via direct correspondence with authors.

## D.3 LEARNING PERFORMANCE

**Environment Configuration** For Atari, we adopt task-specific action space, no-op start with a maximum of 30 steps, episodic life, clip reward, 4 frame-skip, 4 frame-skip, and observation size $84 \times 84$. The ring size is 40. The games of the Atari-5 benchmark suite (Aitchison et al., 2023) includes *BattleZone*, *DoubleDunk*, *NameThisGame*, *Pheonix*, and *Qbert*.

Table 7: Training throughput (FPS) of large-scale experiments in distributed settings, see Fig. 4

| #trainer workers | 8 | 16 | 24 | 32 |
|---|---|---|---|---|
| | Config 1 (SRL) | | | |
| Atari | 161418 | 277214 | 366589 | 453452 |
| DMLab | 180215 | 293325 | 413539 | 493066 |
| SMAC | 14452 | 27037 | 41106 | 53383 |
| GFootball | 89624 | 182725 | 277706 | 337445 |
| | Config 2 (SeedRL style) | | | |
| Atari | 63741 | 89708 | 124682 | 169088 |
| DMLab | 82101 | 183225 | 278620 | 338705 |
| SMAC | 4092 | 7881 | 11055 | 15123 |
| GFootball | 67969 | 131302 | 195204 | 256085 |
| | Config 3 (IMPALA style) | | | |
| Atari | 160820 | 298776 | 345388 | 477904 |
| DMLab | 149457 | 272813 | 332748 | 424252 |
| SMAC | 13359 | 27767 | 40765 | 52725 |
| GFootball | 48320 | 95394 | 148731 | 196305 |

Table 8: Median evaluation score of PPO in Atari across 5 seeds.

| | scale x1 | scale x2 | scale x4 | scale x8 | scale x16 |
|---|---|---|---|---|---|
| BattleZone | 0.0 | 64494.0 | 125439.0 | 108138.0 | 127798.0 |
| DoubleDunk | -1.5 | -0.1 | 22.5 | 22.6 | 22.6 |
| NameThisGame | 19390.4 | 21433.5 | 22202.3 | 23138.2 | 23736.7 |
| Phoenix | 160519.4 | 401654.0 | 425505.6 | 261782.2 | 315078.1 |
| Qbert | 31511.3 | 37704.7 | 45458.3 | 39129.1 | 27532.5 |

For gFootball, we run on four academic scenarios, including *academy_3_vs_1_with_keeper* (3 agents), *academy_counterattack_easy* (10 agents), *academy_counterattack_hard* (10 agents), and *academy_corner* (10 agents). We adopt the "simple115v2" representation, which is a 115-dim vector for each player, and both scoring and checkpoints reward. Besides, rewards are shared across all agents following Yu et al. (2022). The ring size is 8.

**Algorithm Configuration** Per-task scores are shown in Tables 8 to 11. Hyperparameters and resources used in experiments are shown in Table 12. We remark that the hyperparamters of PPO, MAPPO, and VDN follow Yu et al. (2022) and hyperparameters of ApexDQN follow Horgan et al. (2018). We also attempted to benchmark QMix in the gFootball environment, but we found that it performed worse than VDN.

## D.4 COMPUTING RESOURCES

We conduct experiments in the distributed setting using both inline CPU inference (denoted as *CPU Inf.*) and remote GPU inference (denoted as *GPU Inf.*). We fixed the number of actor workers per trainer at 480 for *CPU Inf.*, each with a single environment, and 120 for *GPU Inf.*, each with an environment ring of size 20. *GPU Inf.* uses 8 policy workers along with a trainer, which occupies

Table 9: Median evaluation score of ApexDQN in Atari across 5 seeds.

| | scale x1 | scale x2 | scale x4 | scale x8 | scale x16 |
|---|---|---|---|---|---|
| BattleZone | 47272.0 | 50260.0 | 42167.0 | 37981.0 | 28166.0 |
| DoubleDunk | 22.2 | 0.4 | 17.1 | -0.5 | -1.7 |
| NameThisGame | 12568.1 | 12186.7 | 12476.5 | 12595.2 | 12038.7 |
| Phoenix | 5481.0 | 5334.5 | 5252.4 | 5249.6 | 5071.5 |
| Qbert | 19782.2 | 17198.0 | 9804.9 | 4522.9 | 3088.8 |

Table 10: Median evaluation score (win rate) of VDN in gFootball across 5 seeds.

|        | scale x1 | scale x2 | scale x4 | scale x8 | scale x16 |
|--------|----------|----------|----------|----------|-----------|
| 3v1    | 0.94     | 0.95     | 0.89     | 0.84     | 0.90      |
| Corner | 0.08     | 0.09     | 0.02     | -0.19    | -0.01     |
| CAeasy | -0.22    | -0.17    | -0.20    | -0.22    | -0.23     |
| CAhard | -0.18    | -0.19    | -0.22    | -0.17    | -0.11     |

Table 11: Median evaluation score (win rate) of MAPPO in gFootball across 5 seeds.

|        | scale x1 | scale x2 | scale x4 | scale x8 | scale x16 |
|--------|----------|----------|----------|----------|-----------|
| 3v1    | 0.90     | 0.92     | 0.92     | 0.96     | 0.94      |
| Corner | -0.01    | -0.01    | -0.01    | -0.00    | 0.54      |
| CAeasy | 0.10     | 0.81     | 0.84     | 0.93     | 0.51      |
| CAhard | 0.01     | 0.01     | 0.18     | 0.12     | 0.88      |

two 3090 GPUs. We denote the smallest batch size $0.32M$ used in Baker et al. (2019) as scale x1, which utilizes 4 trainer workers on A100 GPUs in our experiments. Experiments with larger scales duplicate all worker types and increase the batch size by a corresponding factor. Note that the largest scale in Baker et al. (2019) is x4 while the largest scale in our experiments is x8.

### D.5 THE HIDE-AND-SEEK ENVIRONMENT

The hide-and-seek (HnS) environment (Baker et al., 2019) simulates a physical world that hosts 2-6 agents, including 1-3 hiders and 1-3 seekers, and diverse physical objects, including 3-9 boxes, 2 ramps, and randomly placed walls, as shown in Fig. 4a. Agents can manipulate boxes and ramps by moving or locking them, and locked objects can only be unlocked by teammates. At the beginning of each game, there is a preparation phase when seekers are not allowed to move, such that hiders can utilize available objects to build a shelter to defend seekers. After the preparation phase, seekers receive a reward +1 if any of them can discover hiders and -1 otherwise. Rewards received by hiders have opposite signs. The observation, action, and additional environmental details can be found in (Baker et al., 2019).

The four stages emerged across training can be summarized as follows:

1. *Running and chasing*. Seekers learn to run and chase hiders while hiders simply run to escape seekers.
2. *Box lock*. Hiders build a shelter with locked boxes in the preparation phase to defend seekers.
3. *Ramp use*. Seekers learn to utilize ramps to jump over into the shelters built by hiders.
4. *Ramp lock*. Hiders first lock all ramps in the preparation phase and then build a shelter with locked boxes, such that seekers cannot move and utilize ramps.

Table 12: Resources and hyperparameters used in benchmark experiments.

|  | PPO | DQN | MAPPO | VDN |
|---|---|---|---|---|
| discount $\gamma$ | 0.99 | 0.99 | 0.99 | 0.99 |
| GAE $\lambda$ | 0.97 | - | 0.95 | - |
| priority buffer? | False | True | False | False |
| prioritization $\alpha$ | - | 0.6 | - | - |
| prioritization $\beta$ | - | 0.4 | - | - |
| clip ratio | 0.2 | - | 0.2 | - |
| clip value? | True | False | True | False |
| reward transformation? | False | False | False | False |
| value loss | huber ($\delta = 10.0$) | smoothl1 | huber ($\delta = 10.0$) | smoothl1 |
| target update interval | - | 2.5e3 | - | 200 |
| entropy bonus | 0.01 | - | 0.01 | - |
| optimizer | adam | rmsprop | adam | adam |
| learning rate | 5e-4 | 2.5e-4/4 | 5e-4 | 5e-4 |
| optimizer config | PyTorch default | $\alpha = 0.95, \epsilon = 1.5e - 7$ | PyTorch default | $\epsilon = 1e - 5$ |
| buffer warmup size | - | 625 | - | |
| value normalization? | True | False | True | False |
| max gradient norm | 40.0 | 40.0 | 10.0 | 40.0 |
| shared backbone? | True | - | False | - |
| bootstrap steps | 50 | 3 | 50 | 5 |
| data reuse | 5 | - | 5 | - |
| RNN policy? | False | False | True | True |
| rollout length | 80 | 80 | 200 | 200 |
| chunk length | - | - | 10 | 100 |
| network type | DQN | DQN | MLP | MLP |
| hidden size | 512 | 512 | 128 | 128 |
| #rollouts in buffer | 96 | 25000 | 480 | 50000 |
| batch size | 16 | 16 | 80 | 80 |
| actor worker (x1) | 16 | 16 | 80 | 80 |
| policy worker | scale | | max(1,#agents×scale/5) | |
| trainer worker | max(1, scale/8) | | max(1,#agents×scale/48) | |

