# OpenReview forum: "SRL: Scaling Distributed Reinforcement Learning to Over Ten Thousand Cores"
_ICLR.cc/2024/Conference — ICLR 2024 poster_

### Official Review · Reviewer_Wery · 2023-10-23

**Soundness:** 3 good
**Presentation:** 3 good
**Contribution:** 3 good
**Rating:** 8
**Confidence:** 4

**Summary:**

The authors introduce SRL, a system for distributed RL training. This system is built upon a new architecture where *workers* host *task handlers*. There are 3 types of task handlers: actor workers, which execute black box environment programs, policy workers, which handle policy inference, and trainer workers, which compute policy updates. A parameter service is used to store model parameters, and sample and inference streams are defined to send relevant data between the task handlers.

The authors evaluate their system by training on a variety of environments and demonstrating the performance of their system versus baselines in both the single-machine and distributed setting. They also reproduce the hide & seek experiment from OpenAI, which requires a large amount of experience.

**Strengths:**

- The authors describe their system well and the paper is clearly written
- The experiments clearly demonstrate the effectiveness of their system, both in terms of its correctness and scalability
- The design they propose is logical and seems more flexible than comparable approaches.

**Weaknesses:**

- It's unfortunate that the authors were not able to compare with baselines other than RLLIB in the distributed setting, although this is completely understandable given the large-scale distributed training frameworks are closed-source.

**Questions:**

- Can the actor workers run GPU-based environments? These have been shown to have very impressive speed-ups over CPU-based environments [1], and although frameworks like JAX allow for parallelisation across GPUs on the same machine, this is not possible across multiple machines. How does the training speed compare with 8 A100 GPUs (i.e. a single machine with GPU based environments)?


[1] https://github.com/luchris429/purejaxrl/tree/main
[2] https://github.com/RobertTLange/gymnax

---

> ### Author Response · Authors · 2023-11-20
> **Thank you for your valuable and supportive comments!**
>
> We express our gratitude for the reviewer's supportive comments. Our release of SRL primarily aims to empower practitioners to easily customize and deploy RL applications at scale. We appreciate the opportunity provided by the reviewer for publishing our system to benefit the community.
>
> In response to the concern about GPU-based environments, we confirm that SRL supports GPU-based Jax environments across distributed nodes. We evaluated gymnax environments up to 16 A100 GPUs across two nodes and the training throughput linearly scales up.
>
> We remark that SRL was designed as a distributed framework with general environment support. Data in actor workers was assumed to be numpy arrays. As a result, with GPU-based environments, data needs to be copied from GPU to CPU at each environment step. This additional data transmission results in 10x slower environment simulation:
>
> |  gymnax  SpaceInvaders-MinAtar 4096 envs per GPU   | w. GPU-CPU memcpy  |  original |
> |  ----  | ----  | ----  |
> | pure EnvStep per-device FPS |  1.3M | 13.1M |
>
> Consequently, the throughput in the current version of SRL can be sub-optimal. We want to remark that this sub-optimal performance can be resolved by implementing an actor worker that holds Jax/PyTorch GPU tensors in principle, thanks to the modular design of SRL. However, implementing such a customized worker requires tedious engineering efforts to replace every numpy method call to PyTorch/Jax. We are actively implementing this feature and promise to provide full support for GPU environments in our next release.

---

> > ### Comment · Reviewer_Wery · 2023-11-22
> >
> > Thanks for the clarifications. I continue to vote for the acceptance of this paper.

---

### Official Review · Reviewer_bQuB · 2023-11-01

**Soundness:** 3 good
**Presentation:** 3 good
**Contribution:** 3 good
**Rating:** 8
**Confidence:** 3

**Summary:**

In this work the authors implement and evaluate a scalable and efficient framework for highly distributed RL training, known as SRL. Although highly distributed RL training has been studied in previous works they were dominated by industry-led closed-source implementations that yielded very few insights into the correct decompositions and dataflows required to fully utilize the computational hardware on a given system. The authors noted several limitations of existing open-source implementations, stemming from low resource efficiency, inadequate optimizations, or difficult adaptability due to highly coupled algorithm-system interfaces. SRL addresses these issues by presenting a high-level abstraction of the work required to perform RL training into a set of generic workers performing specific tasks connected by different streams of data and coordinated using several services. This abstraction is mapped onto a specific instance of system resources in a flexible manner that facilitates both a large degree of flexibility and resource-aware optimizations to fully exploit the available hardware. The RL-specific components are presented as 3 types of workers to interact with the environment and provide policy updates, 2 streams, and a set of services to perform parameter updating and synchronization. Some optimizations that are specific to RL include the usage of multiple environments to ensure CPU instances can switch between environments without waiting for the next actions to be performed by the policy workers. A wide array of experiments using multiple environments with a mix of CPU and GPU resources demonstrates the utility of SRL to both generate a large number of training frames per second and execute a large number of policy updates per second. Notably, the authors achieve performance that cannot be replicated using any existing open-source RL training frameworks and apply this method to extremely challenging environments, such as hide-and-seek, that require a great deal of data to achieve reasonable performance.

**Strengths:**

- The most noteworthy and interesting aspect of SRL is the demonstrated scalability to support training on an extremely large number of distributed computational resources. Although this level of performance has been noted in other works it was always achieved using RL systems that were not fully released or open-sourced for usage by the general public.
- The flexibility of SRL is evident from the evaluations using a variety of RL algorithms, such as PPO, DQN, and VDN on a number of different system architectures and achieving reasonable performance in all cases. The difference in the necessary computational primitives required by on-vs-off policy methods and single-vs-multi agent environments makes this level of generality challenging and/or overly complicated using many existing RL libraries.
- Well-defined user-friendly extensibility interface appears to be quite attractive for future extensions to scenarios that may not fit neatly into the existing designs typically used common, or the most popular, algorithms used at any particular time.
- The number of evaluations performed on multiple system configurations using a variety of RL environments and incorporating a number of different algorithms is impressive. Figures 3, 4, and 7 do a particularly good job illustrating the level of performance achieved by SRL, comparing that with existing work, and conveying the impact this new level of performance could have on the pace of RL research that may be conducted using open-source software moving forward.
- Many interesting details regarding the implementation specifics weren't clear to me until I read through the appendices. Unfortunately, with a system this wide in scope it is hard to fully understand some aspects without reading through all the additional notes but I was glad the authors provided so many details.

**Weaknesses:**

Major:
- The lack of fair comparison with other large-scale RL training implementations is unfortunate. Though this is by no means the fault of the authors it leaves a level of unknown with regard to a proper comparison of the proposed system against other pre-existing closed-source systems. This is unavoidable and it was good to see the authors attempt to reimplement previous work using their current system to provide some semblance of fairness for comparison.

Minor:
- The text could use a few more passes to fix minor typos in several sections. Nothing major just more editing required.
- Formatting obviously needs to be updated for the code presented in Code 1 and 2 in section 3.4.

**Questions:**

I incorporated all my comments into the strengths and weaknesses sections I have no questions at this time.

---

> ### Author Response · Authors · 2023-11-20
> **Thank you for your thoughtful review and constructive feedback on our paper!**
>
> Thank you for your thoughtful review and constructive feedback on our paper! We appreciate your recognition of our efforts to develop a practical and efficient distributed RL platform for the community. We want to emphasize that this system has been widely adopted within our research group, and we are committed to sharing and refining it further.
>
> Regarding formatting, Code 1 and 2 have been aligned in our revised paper. We will carefully review the paper once more to fix typos.

---

### Official Review · Reviewer_YF9q · 2023-11-01

**Soundness:** 3 good
**Presentation:** 2 fair
**Contribution:** 2 fair
**Rating:** 6
**Confidence:** 4

**Summary:**

This paper proposes a scalable, efficient, and extensible distributed RL framework which can easily scale to ten thousand cores.

**Strengths:**

This paper presents a dataflow abstraction for RL training which allows framework to allocate suitable computing resources in a cluster with heterogeneous hardware. It scales well to ten thousand CPU cores.

**Weaknesses:**

1) MSRL also introduced a data flow abstraction and supported IMPALA and SeedRL distributed architectures. The worker abstraction in SRL is similar to the "fragment" in MSRL. Environment ring is also introduced in EnvPool. Please provide further information about the novelty of SRL.
2) In order to provide a comprehensive understanding of where any gains come from, it would be valuable to include a breakdown ablation study in the evaluation for each optimization and design.
3) It would be beneficial to include MSRL in the experiments to evaluate the gain compared to the state-of-the-art solutions. This could provide deeper insights into where SRL stands in relation to other established techniques.

**Questions:**

1) Please address the above weaknesses.

---

> ### Author Response · Authors · 2023-11-20
> **Author Response**
>
> Thank you for reviewing our paper and we appreciate your comments. Here are our responses to your questions:
>
> > It would be beneficial to include MSRL in the experiments
>
> We want to emphasize that **MSRL results have been included in the experiments already. Please refer to paragraph 2 in Section 4.1**.
>
> We want to further clarify that **we have made our best effort to run the open-source code of MSRL. After we carefully checked the implementation, we have fully confirmed that the MSRL codebase cannot be examined in the same testing scenarios (with a similar scale) as we did**. In MSRL, each environment process requires an independent GPU for policy inference because it uses NCCL as the communication backend. Besides, we didn’t find a multi-learner option in the code. (See [code](https://github.com/mindspore-lab/mindrl/blob/master/mindspore_rl/algorithm/impala/impala_trainer.py).) These implementation choices limit the practical scalability of MSRL. In practice, we can utilize at most 8 CPU cores in the single-machine setting (64 cores for other methods) and at most 64 CPU cores in the distributed setting (700 cores for RLlib and 3200 cores for SRL). We believe this may not become  a fair comparison with such a scalability difference. As a result, we did not include MSRL in our main results.
>
> In addition, as a most sincere attempt to conduct a possible comparison between MSRL and our project, we evaluate SRL under the same setting as the  MSRL paper (Section 6.2 Figure 6(a) in MSRL paper). In this setting, MSRL evaluated the PPO algorithm with 320 Mujoco Halfcheetah environments and 24 Nvidia V100 GPUs for inference. From the numbers in the MSRL paper, MSRL can finish one episode of 1000 steps in 3.85s resulting in 83116 FPS. In our experiment, SRL employs the same number of CPU cores and environment instances, but only need 4 Nvidia 3090 GPUs for inference. Even with 6x fewer GPUs, SRL achieves an overall training throughput of 210165 FPS, which is about 2.52x faster than the reported number of MSRL.
>
> > Strengths: This paper presents a dataflow abstraction for RL training
>
> > Weakness 1: Novelty against MSRL
>
> We respectfully disagree with the reviewer with his/her comments about the contribution and novelty of this paper.
>
> First, we remark that SRL’s contribution is much more than just a proposal of system abstractions. SRL is the first open-source system that can really perform efficient large-scale RL experiments over 15000 cores and reproduce OpenAI’s hide-and-seek project. Besides, SRL offers flexible and extensible interfaces such that users can effortlessly implement a new RL algorithm (see code snippets in Sec3.4 & App.A, various algorithm implementations in Sec.4.2). This paper also performs algorithm evaluation and confirms that insights provided by [1,2] extend to large-scale settings up to 15000 cores, i.e., larger batch sizes are usually beneficial to PPO’s performance. We also want to remark that these experimental and engineering efforts are appreciated by [reviewer Qkzp](https://openreview.net/forum?id=lajn1iROCu&noteId=ITT99z9ky8 ) and [bQuB
> ](https://openreview.net/forum?id=lajn1iROCu&noteId=gPmZgegemV) . We kindly hope the reviewer can consider these aspects when judging this paper.
>
> Second, we have to emphasize that SRL’s abstraction indeed has novelty even compared with the concept proposed by MSRL. Although MSRL’s design concepts can implement both of SEED and IMPALA architectures, it does not provide additional solutions to *overcome their limitations caused by coupling*, as we emphasized and deeply discussed in Sec 2.2. In contrast, workers in SRL are completely decoupled, allowing massive parallelism and efficient resource utilization under large-scale settings.
>
> [1] Bowen Baker, Ingmar Kanitscheider, Todor M. Markov, Yi Wu, Glenn Powell, Bob McGrew, and Igor Mordatch. Emergent tool use from multi-agent autocurricula
>
> [2] Marcin Andrychowicz, Anton Raichuk, Piotr Stanczyk, Manu Orsini, Sertan Girgin, Raphael Marinier, ¨ Leonard Hussenot, Matthieu Geist, Olivier Pietquin, Marcin Michalski, Sylvain Gelly, and Olivier ´ Bachem. What matters for on-policy deep actor-critic methods? A large-scale study.
>
> > Environment ring is also introduced in EnvPool.
>
> We remark that EnvPool runs environments in a thread pool, resembling *parallelism*, while environment ring adopts the idea of *pipelining* between environment simulation and policy inference to reduce idle time.
>
> >In order to provide a comprehensive understanding of where any gains come from,  it would be valuable to include a breakdown ablation study
>
> Please find the breakdown ablation studies in Appendix C. These results are deferred to appendix due to page limits. In ablation studies, we show the benefits from environment ring, GPU inference, and trainer worker prefetching. Moreover, we also study the effect of each worker type and offer practical recommendations on how to configure SRL to achieve the best practical performance.

---

> > ### Comment · Reviewer_YF9q · 2023-11-23
> >
> > I appreciate the author's response to my concerns regarding the comparison with MSRL. It is commendable that the author has put in a significant effort to make the RL framework practical and scalable. As a result, I will increase my score for the paper. Additionally, EnvPool not only increases the parallelism but also offers an async mode to reduce the wait time on both CPU and GPU when the env number is larger than thread number and batch size, which is akin to pipelining between environment and policy inference. Therefore, from the author's response and my understanding, the novelty of SRL lies in its decoupling design and the trainer prefetching.

---

### Official Review · Reviewer_Qkzp · 2023-11-02

**Soundness:** 4 excellent
**Presentation:** 4 excellent
**Contribution:** 3 good
**Rating:** 8
**Confidence:** 3

**Summary:**

The paper introduces SRL, a (really) scalable reinforcement learning framework, designed to parallelize DRL training to large clusters. SRL decomposes DRL training into heterogeneous workers of different types, interconnects them, and schedules them appropriately in order to optimize training throughput. This abstraction is general and enables one to easily implement many different DRL training algorithms. The evaluation demonstrates impressive scalability and training throughput without compromising learning quality.

**Strengths:**

1. SRL is a practical system that seems like it would be of significant benefit to the community. Source code is already (anonymously) available. While the largest scales are likely of interest only to relatively few groups, this capability is nevertheless important, and SRL demonstrates improved performance even at smaller scales (e.g., 32-64 cores).
2. The paper is clear and well-written.
3. The benchmarks cover a number of baselines and consistently show SRL matching or outperforming other systems for DRL training. Results also demonstrate that the system does not compromise learning quality.

**Weaknesses:**

1. While practical, this is primarily "engineering" work and there is not much novelty. (I nevertheless think the practical benefits outweigh this.) In particular, the overall design of the SRL seems very reminiscent of classic task-based programming models from parallel and scientific computing (e.g., Charm++). The paper would benefit from some discussion of this, and it may be a source of inspiration for additional optimizations.
2. The performance optimizations discussed in the main paper, "environment ring" and "trainer pre-fetching", are standard, widely-used optimizations in deep learning frameworks. The environment ring seems to be a case of double-buffering (or pipelining); and many frameworks support prefetching and staging data in advance to the GPU (e.g., while it is not specifically for DRL, the DALI library does this).
3. Performance results lack error bars.

**Questions:**

1. Please clarify or contextualize the novelty of the work (see above).
2. Please add error bars to the performance results.

---

> ### Author Response · Authors · 2023-11-20
> **Thank you for your generous support and insightful comments on our paper!**
>
> Thank you for your generous support and insightful comments on our paper. Our primary aim and focus is indeed to develop a practical and efficient distributed RL platform to benefit the entire community. Your review is extremely encouraging for us. We want to emphasize that this system has been widely adopted within our research group, and we promise to continuously update and maintain the project to really make it useful for everyone.
>
> Regarding the error bar, thanks for the suggestions and we have updated the figures in our paper. We want to clarify that we measured FPS within one hour. The number is stable over such a long time span. As a result, the error bars may look negligible.
>
> Finally, we would still want to discuss the difference between SRL and classical task-oriented programming. Libraries like Charm++ and Ray usually require a driver process (e.g. trainer) to invoke RPCs on remote workers (e.g. actors) to run the training loop. In contrast, SRL does not need a driver and runs independent workers. This can lead to the following differences in the context of RL. First, RPCs may result in additional communication overhead because we need to send requests, while workers in SRL only transfer data among workers to maximize throughput. Second, it is easier to implement customized dataflow with SRL, e.g., controlling multiple agents with different policies, by composing workers and streams. Achieving this with classical task-oriented programming would require the user to change the driver code, which can be tedious.
>
> Hope our reply can address your concerns and thanks again for valuing the efforts of our team for this project.

---

> > ### Comment · Reviewer_Qkzp · 2023-11-22
> > **Response**
> >
> > Thanks for your response and updates. I remain supportive of acceptance.

---

### Meta-Review · Area_Chair_DChp · 2023-12-05

**Metareview:**

This work proposes SRL, a framework for large-scale RL training. The results appear to compare favorably with baselines such as sample factory, while also reproducing the Hide & Seek results from OpenAI. The impact of this may be somewhat limited given that it will still be rather expensive to run, but this at least allows medium sized labs to conduct larger training runs. All reviewers vote for accept and this has been reiterated after the author response.

**Justification For Why Not Higher Score:**

Impact may be limited as 1) it remains expensive to run 2) this area of research is less active nowadays.

**Justification For Why Not Lower Score:**

All reviewers vote accept so it would require a meaningful reason to reject at this point.

---

### Decision · Program_Chairs · 2024-01-16

Accept (poster)